# Efficient Two-Step Adversarial Defense for Deep Neural Networks

## Abstract

In recent years, deep neural networks have demonstrated outstanding performance in many machine learning tasks. However, researchers have discovered that these state-of-the-art models are vulnerable to adversarial examples: legitimate examples added by small perturbations which are unnoticeable to human eyes. Adversarial training, which augments the training data with adversarial examples during the training process, is a well known defense to improve the robustness of the model against adversarial attacks. However, this robustness is only effective to the same attack method used for adversarial training. Madry et al. (2017) suggest that effectiveness of iterative multi-step adversarial attacks and particularly that projected gradient descent (PGD) may be considered the universal first order adversary and applying the adversarial training with PGD implies resistance against many other first order attacks. However, the computational cost of the adversarial training with PGD and other multi-step adversarial examples is much higher than that of the adversarial training with other simpler attack techniques. In this paper, we show how strong adversarial examples can be generated only at a cost similar to that of two runs of the fast gradient sign method (FGSM), allowing defense against adversarial attacks with a robustness level comparable to that of the adversarial training with multi-step adversarial examples. We empirically demonstrate the effectiveness of the proposed two-step defense approach against different attack methods and its improvements over existing defense strategies.

## 1 Introduction

Despite the fact that deep neural networks demonstrate outstanding performance for many machine learning tasks, researchers have found that they are susceptible to attacks by adversarial examples (Szegedy et al. (2014); Goodfellow et al. (2015)). Adversarial examples which are generated by adding crafted perturbations to legitimate input samples are indistinguishable to human eyes. For classification tasks, these perturbations may cause the legitimate samples to be misclassified by the model at the inference time. While there exists no widely agreed conclusion, several studies attempted to explain the underlying causes of the susceptibility of deep neural networks toward adversarial examples. The vulnerability is ascribed to the linearity of the model (Goodfellow et al. (2015)), low flexibility (Fawzi et al. (2018)), or the flatness/curvedness of the decision boundaries (Moosavi-Dezfooli et al. (2017)), but a more general cause is still under research. The recent literature considered two types of threat models: black-box and white-box attacks. In black-box attacks, the attacker is assumed to have no access to the architecture and parameters of the model, whereas in white-box attacks, the attacker has complete access to such information. Several white-box attack methods were proposed (Goodfellow et al. (2015), Papernot et al. (2016a), Su et al. (2017), Carlini & Wagner (2016), Moosavi-Dezfooli et al. (2016)). In response, several defenses have been proposed to mitigate the effect of adversarial attacks. These defenses were developed along three main directions: **(1)** expanding the training data to make the classifier more robustly learn the underlying function, e.g., by adversarial training which augments the training data set with adversarial examples generated by certain attack methods (Szegedy et al. (2014), Goodfellow et al. (2015), Kurakin et al. (2016)); **(2)** modifying the training procedure to reduce the gradients of the model w.r.t. the input such that the classifier becomes more robust to input perturbations, e.g., via input gradient regularization (Ross & Doshi-Velez, 2017), or defensive distillation (Papernot et al., 2016c); and **(3)** using

external models as network add-ons when classifying unseen examples (feature squeezing (Wang et al., 2018), MagNet (Meng & Chen, 2017), and Defense-GAN) (Samangouei et al., 2018)).

Adversarial training, a simple but effective method to improve the robustness of a deep neural network against white-box adversarial attacks, uses the same white-box attack mechanism to generate adversarial examples for augmenting the training data set. However, if the attacker applies a different attack strategy, adversarial training does not work well due to gradient masking (Papernot et al., 2016b). Madry et al. (2017) have suggested the effectiveness of iterative multi-step adversarial attacks. In particular, it was suggested that projected gradient descent (PGD) PGD may be considered the strongest first-order attack so that the adversarial training with PGD can boost the resistance against many other first-order attacks. However, in the literature a large number (e.g. 40) of steps of back propagation are typically used in the iterative attack method of PGD or its closely related variant iterative fast gradient (IFGSM) (Kurakin et al., 2016) to find strong adversarial examples to be used in each adversarial training step, incurring a prohibitively high computational complexity particularly for large DNNs or training datasets.

In this paper, we propose an efficient two-step adversarial defense technique, called e2SAD, to facilitate defense against multiple types of whitebox and blackbox attacks with a quality on a par with the expensive adversarial training using the well-known multi-step attack the iterative fast gradient method (IFGSM) (Kurakin et al., 2016). The first step of e2SAD is similar to the basic adversarial training, where an adversarial example is generated by applying a simple one-step attack method such as the fast gradient sign method (FGSM). Then in the second step, e2SAD attemps to generate a second adversarial example at which the vulnerability of the current model is maximally revealed such that the resulting defense is at the same quality level of the much more expensive IFGSM-based adversarial training. Finally, the two adversarial examples are taken into consideration in the proposed loss function according to which a more robust model is trained, resulting strong defense to both one-step and multi-step iterative attacks with a training time much less less than that of the adversarial training using IFGSM. The main contributions of this paper are as follows:

- We propose a computationally efficient method to generate two adversarial examples per input example while effectively revealing the vulnerability of the learned classifier in the neighborhood of each clean data point;

- We show that by considering the generated adversarial examples as part of a well-designed final loss function, the resulting model is robust to both one-step and iterative white box attacks;

- We further demonstrate that by adopting other techniques in our two-step approach like the use of soft labels and hyper parameter tuning, robust defense against black box attacks can be achieved.

## 2 BACKGROUND

We provide a brief overview of related existing attacks and defense methods, part of which will be also used to compare with the proposed e2SAD approach.

**Attack Models and Algorithms.** The goal of all attack models is to find a perturbation $\delta$ to be added to a clean input $x \in \mathbb{R}^d$, resulting in an adversarial example $x_{adv} = x + \delta$ which may potentially lead to misclassification of the classifier. Typically, the noise level of the perturbation is constrained by the $\ell_\infty$ ball denoted by $\varepsilon$ to make sure that the perturbation is sufficiently small. Based on the amount of information the attacker knows, there are two threat levels as follows:

1. White box: the attacker has full information about the model including its architecture and parameters such that it is possible craft adversarial examples using techniques such as gradient based attacks to specifically target the model;

2. Black box: the attacker has no knowledge about the architecture and parameters of the model. Neither is the attacker able to query the model. Adversarial examples can be generated using a substitute model which is a white-box to the attacker.

## 2.1 WHITE BOX ATTACKS

**The Fast Gradient Sign Method (FGSM).** Given a clean input $x$ and its corresponding true label y, FGSM perturbs x by (Goodfellow et al., 2015):

$$x^{adv} = x + \varepsilon \cdot sign(\nabla_x J(\theta, x, y)) \tag{1}$$

where $J(\theta, x, y)$ is the loss function and $\varepsilon \in [0, 1]$ is a constant value used to constrain the noise level of the perturbation.

**Iterative Fast Gradient Sign Method (IFGSM) and PGD** IFGSM attack generates adversarial examples by iteratively applying FGSM attack multiple, say $N$, times to a clean input $x$ with a small constant $a$ (Kurakin et al., 2016)

$$x_k^{adv} = clamp_{x,\varepsilon} \left( x_{k-1}^{adv} + a \cdot sign(\nabla_{x_{k-1}^{adv}} J(\theta, x_{k-1}^{adv}, y)) \right), \ x_0^{adv} = x, \ k = [1, \cdots, N]. \tag{2}$$

In our implementation, we set $a = \frac{\varepsilon}{N}$. Typically, each component of the input vector, e.g. a pixel, is normalized to be within [0, 1]. The function $clamp_{x,\varepsilon}$ is an elementwise clipping function which clips each element $x_i$ of input $x$ into the range of $[max(0, x_i - \varepsilon), min(1, x_i + \varepsilon)]$.

Projected gradient descent (PGD) is a closely related variant of IFGSM. Typically, PGD first randomly picks a point within a confined small ball around each clean input and then applies the multistep IFGSM to generate adversarial examples for that clean input.

## 2.2 REPRESENTATIVE EXISTING DEFENSE METHODS

**Adversarial Training.** This is a popular defense approach which augments the training dataset with adversarial examples (Goodfellow et al. (2015), Kurakin et al. (2016)). In our implementation, we adopt the adversarial training equation proposed in (Goodfellow et al., 2015) as the loss function

$$J_{adv(\theta,x,y)} = \alpha \cdot J(\theta, x, y) + (1 - \alpha) \cdot J(\theta, x^{adv}, y), \tag{3}$$

where $\alpha$ is a constant specifying the relative importance of the adversarial examples. In our latter comparison, we choose two methods, FGSM and IFGSM, for generating the adversarial examples.

**Minmax.** There exist defense methods (Huang et al. (2015); Madry et al. (2017)) which view the process of training a robust model as solving a minmax optimization problem

$$\arg\min_{\theta} \mathbb{E}_{(x,y) \sim \mathcal{D}} \left[ \max_{\delta} J(\theta, x + \delta, y) \right], \tag{4}$$

where $\mathcal{D}$ is the underlying training data distribution, $J(\theta, x, y)$ is the loss function, and $\theta$ is the parameters of the model. In (Huang et al. (2015)) and Madry et al. (2017), the maximization with respect to $\delta$ is approximated by a specific attack method, for example, by PGD Madry et al. (2017).

Hamm (2018) proposes a new approach which can instead of targeting the saddle points like previous methods, find the true optimal solution of the minmax optimization problem. Hamm (2018) chooses FGSM to approximate the inner maximization step and for the outer minimization step, instead of plugging the adversarial version of the clean data directly and solving the optimization problem, changes the minimization objective function to

$$\arg\min_{\theta} \left[ J(\theta, x^{adv}, y) + \frac{\varepsilon N}{2} \left\| \frac{\partial J(\theta, x^{adv}, y)}{\partial x^{adv}} \right\| \right], \tag{5}$$

where $\varepsilon$ is a constant restricting the level of input perturbation, and $N$ is the number of the training examples in each minibatch.

# 3 METHODS

## 3.1 ADVERSARIAL TRAINING

Adversarial training, which augments the training dataset with adversarial examples during the training process, has been shown to increase the robustness of the model against white box attacks when the attack method used to generate the augmented training set is the same as the method used by the attacker. However, if the attacker uses a different attack strategy to apply the white box attack, adversarial training does not perform well. For example, adversarial training using one-step FGSM can not improve the robustness of the model against multi-step attacks such as IFGSM and PGD. However, compared to adversarial training using IFGSM or PGD, adversarial training using one-step FGSM takes much less time for the training process since it takes only one step of back propagation to generate adversarial example during each training iteration. Madry et al. (2017) suggest that PGD, one particular type of multi-step iterative adversarial attack, is the strongest universal first-order adversary. It is also suggested that the model trained by the adversarial training with PGD is robust against both PGD and one-step FGSM, however at the expenses of multiple steps of back propagation per a clean training data point.

## 3.2 PROPOSED EFFICIENT TWO-STEP ADVERSARIAL DEFENSE (E2SAD)

Our ojective is to develop a defense method with a cost similar to that of FGSM adversarial training while being robust to both FGSM and multiple-step attacks such as IFGSM.

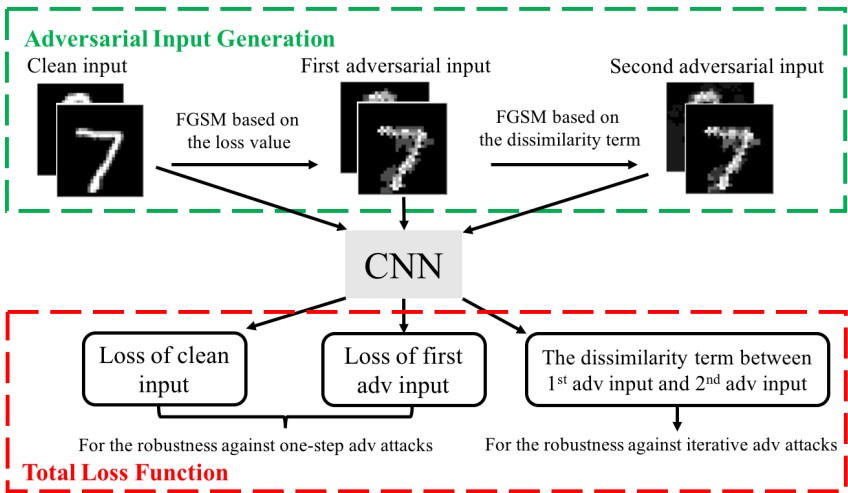

Figure 1: e2SAD: the proposed efficient two-step adversarial defense method.

A shown in Figure 1, the proposed efficient two-step adversarial defense (e2SAD) approach takes only two steps of back propagation to find adversarial examples. First, for a given input we define the *categorical distribution* of the model as the vector of probabilities the model outputs, where each component of the vector representing the probability for the input to be in the corresponding class. At the first step of e2SAD, a one-step attack method such as FGSM is applied to find the first adversarial example per each clean input. At the second step, within the neighborhood of the first adversarial example, the input point whose categorical distribution is most different from that of the first adversarial point is selected as the second adversarial example. For each clean input, these two generated adversarial examples are considered in the final loss function for training. The loss function consists of three terms: the loss of the original clean inputs, the loss of the adversarial examples generated at the first step, and the dissimilarity in categorical distribution of all pairs of the corresponding first and second adversarial examples. It is worth noting that the two-step e2SAD approach is structured in a particular way such that it may provide strong defense against both one-step and multiple-step attacks, as detailed below.

### 3.2.1 ROBUSTNESS AGAINST ONE-STEP ADVERSARIAL ATTACKS

The main objective of the first step of e2SAD is to find a highly vulnerable neighborhood immediately around each clean training data point such that the trained model can be made robust to one-step gradient-based attacks. In so doing, we simply apply a one-step attack method such as FGSM to maximize the loss around around each clean input $x_i$ to generate the first adversarial example $x_i^{adv}$

$$x_i^{adv} = x_i + \varepsilon_1 \cdot sign(\Delta_{x_i} J(\theta, x_i, y)), \tag{6}$$

where $\varepsilon_1$ is a constant chosen step size. We include the loss of this adversarial example in the final loss function (8), discussed in detail in the next subsection. Essentially, by doing so, the first term of (8) guides the training process to reduce the losses of both $x_i$ and $x_i^{adv}$, acting as a mechanism for defending one-step adversarial attacks.

### 3.2.2 ROBUSTNESS AGAINST ITERATIVE ADVERSARIAL ATTACKS

As discussed earlier, compared to one-step adversarial attacks iterative multi-step attacks can be much stronger as they search the neighborhood of a clean data point more exhaustively, which in turns makes the adversarial training using iterative adversarial attacks a stronger defense. At the second step of e2SAD, our goal is to efficiently defend against multi-step attacks by using only one extra step of computation. As such, the key challenge here is to find a second adversarial example $\tilde{x}_i^{adv}$ which is close to $x_i^{adv}$ and can effectively reveal the vulnerability of the model in a way similar to expensive multi-step attacks.

In a multi-step attack method such as IFGSM or PGD, each adversarial example in the iterative process is typically found by perturbing the preceding adversarial example to maximize its loss, where the loss, for example, may be described using the cross entropy based on either the hard or soft label. Despite this common practice, we argue that a more appropriate approach is to instead constrain the training process such that a level of *similarity* (or uniformity) in the prediction of the trained model is maintained in the neighborhood of each clean input $x_i$. It is important to note that *maintaining similarity of prediction* and *minimizing the loss* may be correlated but are not necessarily identical objectives; the latter attempts to ensure that predictions made in some neighborhood of the input individually have low loss without specifically constraining these predictions to be similar to each other. Nevertheless, we believe that the objective of maintaining similarity of prediction is more relevant as far as adversarial defense is concerned as it may lead to a well-regularized decision boundary around each $x_i$.

With the above understanding, at the second step of e2SAD, we attempt to find the second adversarial example $\tilde{x}_i^{adv}$ whose categorical distribution is maximally different from that of the first adversarial example $x_i^{adv}$ in the neighborhood of $x_i^{adv}$. The dissimilarity in categorical distribution between these two points is measured by cross entropy (CE). To locate $\tilde{x}_i^{adv}$, FGSM is used as a one-step optimization method to maximize the CE-based dissimilarity measure

$$\tilde{x}_i^{adv} = x_i^{adv} + \varepsilon_2 \cdot sign\left(\nabla_{\tilde{x}_i^{adv}} CE(f(x_i^{adv}, \theta), f(\tilde{x}_i^{adv}, \theta))|_{\tilde{x}_i^{adv} = x_i^{adv}}\right), \tag{7}$$

where $\varepsilon_2$ is the step size, and the gradient of the CE-based dissimilarity is evaluated at $x_i^{adv}$.

The reason for using categorical distribution as the measure of dissimilarity to find the second adversarial point is as follows. First, note that the value of loss for a model prediction does not fully indicate whether the prediction is a misclassification or not. To see this, consider a simple classification task with three classes. Assume that the true class labels for two different inputs are both the first class, and the corresponding categorical distributions are $[0.45, 0.55, 0]$ and $[0.4, 0.3, 0.3]$, respectively. Let us further assume that one-hot encoding is conventionally used for the labels. In this case, the model misclassifies the first input while correctly classifies the second. However, this happens even when the loss of the first input is lower than that of the second input.

Figure 2 shows how the choice of the optimization objective may influence the generation of the second adversarial example for an illustrative three-class classification problem. The probabilities of three classes predicted by a trained model for a set of inputs are illustrated using the green, red, and purple curves, respectively. Accordingly, the one-hot encoding loss as a function of the input is shown by the blue curve. The cross entropy of categorical distribution between each input and

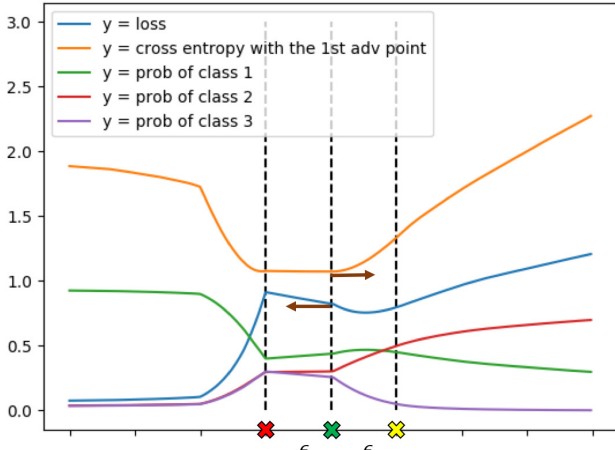

Figure 2: Using the loss and cross-entropy dissimilarity at the second step of e2SAD for a three-class classification problem. Both the clean input and first adversarial example belong to class 1. Green cross: the 1st adversarial point; Red cross: the 2nd adversarial point found by applying FGSM on the loss value; Yellow cross: the 2nd adversarial point found by applying FGSM on the cross entropy dissimilarity with the 1st adversarial point. Only the yellow cross corresponds to a misclassification.

the first adversarial example (green cross) is shown by the orange curve. Note that both the clean input and first adversarial example belong to class 1 in this setup. Starting from the green cross, maximizing the loss using FGSM produces the red cross as the second adversarial example. In comparison, using the CE dissimilarity measure as the objective function leads to the yellow cross. While having the highest loss, the red cross is correctly classified by the model. On the other hand, misclassification happens at the yellow cross which is found based on the CE dissimilarity measure, suggesting its effectiveness in finding stronger adversarial points.

### 3.2.3 THE FINAL LOSS FUNCTION

Based the two adversarial examples generated at the two steps of e2SAD, we design the loss function used for training the final model as follows. For a mini batch $X$ of $m$ clean examples $\{x_1, \cdots, x_m\}$ and the corresponding mini batch of the first set of adversarial examples $X^{adv} = \{x_1^{adv}, \cdots, x_m^{adv}\}$ generated at the first step of e2SAD, the total loss function is given by

$$\arg\min_{\theta} \left[ \alpha \cdot J(\theta, X, Y), +(1-\alpha) \cdot J(\theta, X^{adv}, Y) + \lambda \frac{1}{m} \sum_{i=1}^{m} D(f(x_i^{adv}, \theta), f(\tilde{x}_i^{adv}, \theta)) \right], \quad (8)$$

where $\theta$ is the parameters of the model, each $\tilde{x}_i^{adv}$ is the second adversarial example which has the maximally different categorical distribution from the corresponding first adversarial example $x_i^{adv}$, $f(x, \theta)$ indicates the categorical distribution output function of the model for input $x$, and $D$ is the cross-entropy dissimilarity measure.

### 3.3 THE TRAINING PROCESS

We adopt label smoothing (Szegedy et al., 2014) for the training process. Here, instead of using hard labels (one-hot labels) for each cross-entropy loss, we employ the so-called soft labels which assign the correct class a target probability of $1 - \delta$ and divide the remaining $\delta$ probability mass uniformly among the incorrect classes. We have found that the use of label smoothing in e2SAD leads to better performance.

The overall training algorithm of the proposed e2SAD approach is summarized in Algorithm 1. The hyperparameters $\alpha$ and $\lambda$ specify the weights for the losses of the clean and first set of adversarial

inputs and those for the dissimilarity between each pair of the first and second adversarial inputs, respectively. While the first two terms in the final loss function target the defense against one-step adversarial attacks, the last term mainly plays the role of defending multi-step attacks. In practice, $\alpha$ and $\lambda$ shall be properly chosen to balance between these two different defense needs.

---

**Algorithm 1** The proposed two-step adversarial defense: e2SAD

---

**Input:** training dataset $(X, Y)$; Initial model parameter $\theta$; batch size: $m$; hyperparameters $\alpha, \lambda, \varepsilon_1, \varepsilon_2$

**Output:** Trained model parameter $\theta$

1: **for** each minibatch t **do**
2:     **for** each $(x_i, y_i)$ in the current batch **do**
3:         $x_i^{adv} \leftarrow x_i + \varepsilon_1 \cdot sign(\nabla_{x_i} CE(y_i, f(x_i, \theta)))$
4:         $\tilde{x}_i^{adv} \leftarrow x_i^{adv} + \varepsilon_2 \cdot sign\left(\nabla_{\tilde{x}_i{}^{adv}} CE(f(x_i^{adv}, \theta), f(\tilde{x}_i{}^{adv}, \theta))|_{\tilde{x}_i{}^{adv} = x_i^{adv}}\right)$
5:     **end for**
6:     $L_{adv} = \alpha \cdot J(\theta, X_t, Y_t) + (1 - \alpha) \cdot J(\theta, X_t^{adv}, Y_t)$
7:     $L_d = \frac{1}{m}\lambda \sum_{i=1}^{m} D(f(x_i^{adv}, \theta), f(\tilde{x}_i{}^{adv}, \theta))$
8:     $L_{total} = L_{adv} + L_d$
9:     Update $\theta$ using backpropogation based on $L_{total}$
10: **end for**
11: Return $\theta$

---

We visually show the two-step e2SAD adversarial example generation process and the loss surfaces of four different models for a minibatch of 128 clean images from the MNIST handwritten digits dataset (LeCun et al., 1998) in Figure 3 and Figure 4 of the Appendix, respectively, to demonstrate the effectiveness of e2SAD.

## 4 EXPERIMENTAL RESULTS

We compare the proposed e2SAD method with two widely adopted techniques in the literature: adversarial training using single-step FGSM (Goodfellow et al., 2015) and the adversarial training using multi-step IFGSM. We also report our experience on the minimax adversarial defense method proposed in (Hamm, 2018). We adopt the widely used the MNIST handwritten digits dataset (LeCun et al., 1998) and the Street View House Numbers (SVHN) Dataset (Netzer et al., 2011) as benchmarks.

### 4.1 RESULTS ON THE MNIST HANDWRITTEN DIGITS DATASET

MNIST consists 60,000 training images and 10,000 testing images, where each pixel value is normalized to be within $[0, 1]$. The adversarial attacks considered are:

- White-box attacks with FGSM under different noise levels: $\varepsilon = 0.3, 0.4$.

- White-box attacks with IFGSM under the fixed noise level of $\varepsilon = 0.3$ with different numbers of steps: $k = 10, 30$.

- Black-box attacks from three substitute models: the naturally trained model (i.e. the one trained using only the clean inputs without any additional defense strategy), one trained with FGSM adversaries under the noise level of $\varepsilon = 0.3$, and one trained with IFGSM adversaries under the total noise level of $\varepsilon = 0.3$ and step size of 0.01 ($k = 30$). With respect to these substitute models, IFGSM with $\varepsilon = 0.3$ and $k = 30$ is used to generate adversarial examples, which are then employed to attack each of the targeted models.

All CNNs we use consist of two convolutional layers with 32 and 64 filters respectively, each of which is followed by a $2 \times 2$ max-pooling layer and ReLU activation function, and a fully connected layer of $1,024$ neurons. The configuration of the CNNs is summarized in Table 4 in the Appendix.

4.1.1 RESULTS ON WHITE-BOX ATTACKS

For our proposed e2SAD method, we set the hyperparameters in the training Algorithm 1 as: $\alpha = 0.6$, $\lambda = 0.1$, $\varepsilon_1 = 0.3$, and $\varepsilon_2 = 0.1$. To increase the searching ability of the second step of e2SAD, we do not clamp the second adversarial point to be within a norm ball around the clean data point. All models are trained on MNIST for 30,000 iterations with the batch size of 256.

**Comparison with adversarial training**  The performances of different models under various white-box attacks are shown in Table 1. It can be seen that each model reaches the accuracy of over 99% on the clean dataset. The baseline model trained naturally shows no defense ability towards both FGSM and IFGSM adversaries while other three models demonstrate different levels of defense. The model obtained via FGSM adversarial training maintains a very high accuracy under FGSM attacks with different noise levels. However, FGSM adversarial training can only defend FGSM attacks while shows no defense ability against IFGSM attacks of any step number. IFGSM adversarial training performs well under IFGSM adversaries and also shows robustness against FGSM attacks. However, the defense ability drops fast when the noise level $\varepsilon$ increases in the case of FGSM attacks. Specifically, the accuracy can drop by almost 14% under the FGSM attacks when the noise level increases to $\varepsilon = 0.4$. Note that it makes 30 steps to generate IFGSM adversarial examples in each training iteration, leading to the high cost of the considered IFGSM adversarial training.

Table 1: Accuracy of different models under white-box adversarial attacks evaluated using MNIST. Rows are the attacks where "Clean Data" in the first row means no attack. Columns report the accuracy of different defense solutions where "Natural" means the baseline model without any additional defense strategy, and "FGSM Adv. Train" and "IFGSM Adv. Train" mean the adversarial training with FGSM and IFGSM, respectively. "H"("S") stands for use of one-hot hard (soft) labels when training the defense model.

| Attack | | Label | Natural | FGSM Adv. Train | | IFGSM Adv. Train | e2SAD |
|---|---|---|---|---|---|---|---|
| | | | | $\varepsilon = 0.3$ | $\varepsilon = 0.4$ | $\varepsilon = 0.3, k = 30$ | |
| Clean Data | | H | 0.9942 | 0.9938 | 0.9921 | 0.9913 | 0.9932 |
| | | S | 0.9952 | 0.9943 | 0.9941 | 0.9913 | |
| FGSM | $\varepsilon = 0.3$ | H | 0.1741 | 0.9768 | 0.9658 | 0.9519 | 0.9641 |
| | | S | 0.4256 | 0.9846 | 0.9919 | 0.9595 | |
| | $\varepsilon = 0.4$ | H | 0.1027 | 0.9136 | 0.9732 | 0.8152 | 0.9499 |
| | | S | 0.2146 | 0.9374 | 0.9914 | 0.9012 | |
| IFGSM | k=10 | H | 0.0001 | 0.0856 | 0.2108 | 0.9336 | 0.8687 |
| | | S | 0.1019 | 0.1166 | 0.0101 | 0.9422 | |
| | k=30 | H | 0 | 0.082 | 0.1968 | 0.9325 | 0.8633 |
| | | S | 0.093 | 0.0966 | 0.0065 | 0.9412 | |

Among all models considered, the proposed e2SAD method produces the highest accuracy for both the clean data and FGSM attacks at different noise levels. Under IFGSM attacks e2SAD significantly outperforms the FGSM adversarial training, demonstrating the effectiveness of the proposed two-step approach's generalization capability with respect to defense against strong multi-step attacks. Compared with the adversarial training using IFGSM, e2SAD offers stronger defense against FGSM attacks while maintaining a good robustness against IFGSM attacks. Note that these are achieved using only two steps of gradient calculation in each training iteration, presenting a significant reduction of computational cost compared with the IFGSM adversarial training, which performs 30 steps of gradient computation.

Label smoothing is adopted in e2SAD and it is shown to be effective in helping the trained model generalize well. In our experiments, we set the probability for the correct label to 0.75 and the one for all other incorrect labels to 0.25. Table 1 shows that label smoothing also improves the performance of the traditional adversarial training under some circumstances, but not significantly.

Table 2: Accuracy of different models under black-box IFGSM attacks evaluated using MNIST.

| Substitude Model | Label | Natural | FGSM Adv. Train | | IFGSM Adv.Train | e2SAD |
|---|---|---|---|---|---|---|
| | | | $\varepsilon = 0.3$ | $\varepsilon = 0.4$ | $\varepsilon = 0.3, k = 30$ | |
| Natural Model (H) | H | 0 | 0.9083 | 0.9158 | 0.9668 | 0.868 |
| | S | 0.1376 | 0.7887 | 0.8963 | 0.9641 | |
| FGSM (H) $\varepsilon = 0.3$ | H | 0.9127 | 0.082 | 0.8945 | 0.967 | 0.9422 |
| | S | 0.9083 | 0.7581 | 0.8214 | 0.9671 | |
| IFGSM (H) $\varepsilon = 0.3, k = 30$ | H | 0.9163 | 0.9319 | 0.9148 | 0.9324 | 0.8886 |
| | S | 0.885 | 0.768 | 0.8482 | 0.9574 | |
| Natural Model (S) | H | 0.9024 | 0.9742 | 0.9674 | 0.9838 | 0.9719 |
| | S | 0.0929 | 0.8485 | 0.8963 | 0.9847 | |
| FGSM (S) $\varepsilon = 0.3$ | H | 0.9777 | 0.9708 | 0.9543 | 0.9825 | 0.9698 |
| | S | 0.9745 | 0.0966 | 0.7658 | 0.9832 | |
| IFGSM (S) $\varepsilon = 0.3, k = 30$ | H | 0.939 | 0.952 | 0.9464 | 0.9578 | 0.9525 |
| | S | 0.9435 | 0.9476 | 0.9481 | 0.9412 | |

**Comparison with the minimax adversarial defense** We also implemented the minimax adversarial defense method proposed in (Hamm, 2018) with a minor modification that the model is trained using a mixture of clean and adversarial examples to achieve better performance. Our results show that the trained model is very robust against FGSM attacks, however, shows no defense against IFGSM attacks.

### 4.1.2 RESULTS ON BLACK-BOX ATTACKS

In Table 2, we consider how adversarial examples generated by applying IFGSM to a substitute model may attack a different model. The rows of the table are the considered substitute models: "Natural model" is again the baseline model without any additional defense strategy; "FGSM $\varepsilon = 0.3$" is the model obtained via FGSM adversarial training with the setting $\varepsilon = 0.3$; "IFGSM $\varepsilon = 0.3, k = 30$" is the model obtained via IFGSM adversarial training with the setting $\varepsilon = 0.3, k = 30$. The substitution models are trained using hard labels ("H") and label smoothing ("S"), then *attacked by IFGSM* with the setting ($\varepsilon = 0.3, k = 30$) for generating adversarial examples. The adversarial examples generated from the substitute models are used to attack the four models shown in the columns of the table: "Natural" is the baseline model; "FGSM Adv. Train" and "IFGSM Adv. Train" are models trained by the FGSM and IFGSM adversarial training using the settings specified in the table, respectively; "e2SAD" is the proposed model. The models under attack are trained using both hard and label smoothing except for e2SAD which is based on label smoothing only. Note that in Table 2, white-box attacks are resulted when the substitute model and the one under attack are identical, and all other combinations correspond to black-box attacks.

Table 2 demonstrates that the proposed e2SAD approach delivers a well-balanced defense against black-box IFGSM attacks from all three substitute models with an accuracy of nearly 90% or higher. There are several cases under which the natural training (baseline) and FGSM adversarial training have a poor performance. In all cases, e2SAD either noticeably outperforms both the natural training and FGSM adversarial training or produce a fairly close performance. Compared with the models trained with the 30-steps IFGSM adversarial training, e2SAD is still very competitive particularly given the fact that only two-steps of gradient computation are performed at each training iteration.

### 4.2 RESULTS ON THE STREET VIEW HOUSE NUMBERS (SVHN) DATASET

The Street View House Numbers (SVHN) dataset ( (Netzer et al., 2011)) consists of a training set of 73,257 digits and a testing set of 26,032 digits obtained from house numbers in Google Street View images, representing a significantly harder real-world dataset compared to MNIST. We process the SVHN dataset by removing the mean and normalizing the pixel values with the standard deviation of all pixels in each image so that the normalized pixel values are within [-1, 1].

We train three different models with the CNN configuration summarized in Table 5 in the Appendix and compare their performances under the scenario of white box attacks. All models are trained for 20 epochs with the following setup

- FGSM-based adversarial training: {Batch size = 256, optimizer=AdamOptimizer with learning rate 0.001, $\alpha = 0.6$, $\varepsilon = 24/255$}
- IFGSM-based adversarial training: {Batch size = 256, optimizer=AdamOptimizer with learning rate 0.001, $\alpha = 0.6$, $\varepsilon = 24/255$, attack steps=10}
- e2SAD: {Batch size = 256, optimizer=AdamOptimizer with learning rate 0.001, $\alpha = 0.6$, $\lambda = 0.3$, $\varepsilon_1 = 24/255$, $\varepsilon_2 = 8/255$, label smoothing with correct class probability of 0.75}

The performances of the various models on this much harder SVHN dataset are summarized in Table 3. It turns out that e2SAD outperforms all other models in this case. More specifically, the baseline (natural) model shows no defense to any attack. e2SAD attains a significantly stronger robustness against the iterative IFGSM white-box attacks compared with the FGSM adversarial training, which shows no defense to such attacks. Furthermore, compared with the expensive IFGSM adversarial training, e2SAD offers a much stronger defense against the one-step FGSM attacks. This fact may be attributed to the particular two-step structure of e2SAD, which is geared towards defending both one-step and multi-step adversarial attacks.

Table 3: Test accuracy of different models under white-box adversarial attacks evaluated using SVHN. The e2SAD models are trained with soft labels while all other models are trained with hard labels. The definitions of all variables are identical to those used in Table 1.

| Attack | | Natural | FGSM Adv. Train $\varepsilon = 24/255$ | IFGSM Adv. Train $\varepsilon = 24/255, k = 10$ | e2SAD |
|---|---|---|---|---|---|
| Clean data | | 0.9006 | 0.9119 | 0.9001 | 0.9236 |
| FGSM | $\varepsilon = 24/255$ | 0.1962 | 0.7842 | 0.4289 | 0.7881 |
| IFGSM | $\varepsilon = 24/255$, k=10 | 0.0628 | 0.0455 | 0.3228 | 0.3328 |
| | $\varepsilon = 24/255$, k=20 | 0.0602 | 0.0402 | 0.3165 | 0.4020 |
| | $\varepsilon = 24/255$, k=30 | 0.0593 | 0.0385 | 0.3146 | 0.3868 |

## 5 CONCLUSION

We have aimed to improve the robustness of deep neural networks by presenting an efficient two-step adversarial defense technique e2SAD, particularly w.r.t to strong iterative multi-step attacks. This objective is achieved by finding a combination of two adversarial points to best reveal the vulnerability of the model around each clean input. In particular, we have demonstrated that using a dissimilarity measure between the first and second adversarial examples we are able to appropriately locate the second adversary in a way such that including both types of adversaries in the final training loss function leads to improved robustness against multi-step adversarial attacks. We have demonstrated the effectiveness of e2SAD in terms of defense against while-box one-step FGSM and multi-step IFGSM attacks and black-box IFGSM attacks under various settings.

e2SAD provides a general mechanism for defending both one-step and multiple attacks and for balancing between these two defense needs, the latter of which can be achieved by properly tuning the corresponding weight hyperparameters in the training loss function. In the future work, we will explore hyperparameter tuning and other new techniques to provide a more balanced or further improved defense quality for a wider range of white and black box attacks.

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

## A APPENDIX

### A.1 CNN MODEL CONFIGURATIONS

| Layer Type | Details |
|---|---|
| ReLU Convolutional | 32 filters (5*5, stride 1, padding 2) |
| Max Pooling | 2*2 |
| ReLU Convolutional | 64 filters (5*5, stride 1, padding 2) |
| Max Pooling | 2*2 |
| ReLU Fully Connect | 1024 units |
| Fully Connect | 10 units |
| Softmax | 10 units |

Table 4: CNN model configuration for the MNIST dataset.

| Layer Type | Details |
|---|---|
| ReLU Convolutional | 32 filters (5*5, stride 1, padding 0) |
| ReLU Convolutional | 64 filters (5*5, stride 1, padding 0) |
| Max Pooling | 2*2 |
| ReLU Convolutional | 128 filters (3*3, stride 1, padding 0) |
| Max Pooling | 2*2 |
| ReLU Fully Connect | 512 units |
| Fully Connect | 10 units |
| Softmax | 10 units |

Table 5: Model configuration for the SVHN dataset.

### A.2 VISUALIZATION OF E2SAD GENERATED ADVERSARIAL EXAMPLE

To demonstrate the two-step adversarial generation process of e2SAD, we consider a minibatch of 128 clean images from the MNIST handwritten digits dataset (LeCun et al., 1998). We apply e2SAD to find the first and second adversarial examples for each clean image $x_i$ in the batch. To help visualize the loss surface of the model around this minibatch, which may be explored by IFGSM attacks in a two-dimensional input space, we identify a search direction $g_1 = sign(x_i^{adv,IF} - x_i)$, where $x_i^{adv,IF}$ is the adversary for $x_i$ found by IFGSM. We define a second search direction $g_2$ to be orthogonal to $g_1$. Then around each $x_i$, we generate a set of perturbed images along $g_1$ and $g_2$: $X_p = \{x_i + t_1 \cdot g_1 + t_2 \cdot g_2\}$, $t_1, t_2 \in [0, 0.4]$. $t_1$ and $t_2$ are chosen to be the two lateral axes in Figure 3. Here the loss is defined as the cross entropy loss based on hard target labels. The mesh loss surface shows the loss of the model summed over the perturbed images for the entire minibatch as a function of $t_1$ and $t_2$. The blue dot at $(0, 0)$ location is the loss of the minibatch of clean images. The red line starting from this blue point illustrates the two-step e2SAD adversarial searching direction. The second and third blue points on the red line show the losses summed over the first and second sets of adversarial examples, respectively, generated by e2SAD for this minibatch of clean images. The locations of these two points are projected on the $t_1$ and $t_2$ coordinates for visualization. In this case, at the second step e2SAD is able to identify an effective set adversarial examples with a cost further increased from the first set, suggesting its effectiveness in defending both one-step and multi-step adversarial attacks.

### A.3 VISUALIZATION OF THE LOSS SURFACES OF FOUR DIFFERENT MODELS

We visualize the loss surfaces of different models to shed light on the potential defense capabilities of these models with respect to both one-step FGSM attacks and multi-step IFGSM attacks in Figure 4a and Figure 4b, respectively. Here, the baseline model again is only trained with the clean data and with no additional defense strategy; "FGSM Adv. Train" is the model is trained by adversarial training with adversaries generated from FGSM ($\varepsilon = 0.3$); "IFGSM Adv. Train" is the model trained by adversarial training with adversaries generated from IFGSM ($\varepsilon = 0.3, k = 30$); And

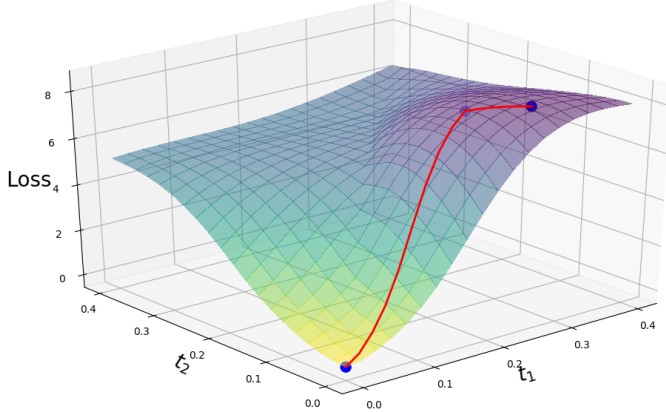

Figure 3: The two-step adversarial example generation process of e2SAD for a minibatch of 128 MNIST images.

e2SAD is the proposed approach with the setting ($\theta = 0.2, \lambda = 0.1, \varepsilon = 0.3, \varepsilon = 0.1$). All models are trained using a total number of 30,000 mini-batches of 256 images each over the MNIST dataset.

Figure 4a and Figure 4b illustrate the loss surface of each model in the input space, which may be viewed by FGSM and IFGSM attacks, respectively, when they generate adversarial examples. To make visualizations possible in a reduced 2-dimensional input space, we take the approach adopted in Figure 3. For example, in the case of Figure 4a, we identify a search direction $g_1 = sign(x_i^{adv,FGSM} - x_i)$, where $x_i^{adv,FGSM}$ is the adversary for each clean image $x_i$ found by the FGSM attack. We define a second search direction $g_2$ to be orthogonal to $g_1$. Then around each $x_i$, we generate a set of perturbed images along $g_1$ and $g_2$: $X_p = \{x_i + t_1 \cdot g_1 + t_2 \cdot g_2\}$, $t_1, t_2 \in [0, 0.4]$. $t_1$ and $t_2$ are again chosen to be the two lateral axes in Figure 4a as in Figure 3. The mesh loss surface of a model shows the loss summed over the perturbed images for the entire MNIST dataset as a function of $t_1$ and $t_2$. Again, the value at $(0, 0)$ location is the loss of all (MNIST) clean images. The same visualization approach is taken in Figure 4b with the difference that the two search directions are defined by the adversary found by the IFGSM attack for each clean image.

In both figures, it can be observed that the loss surface of the e2SAD model is the flattest one with the lowest average value within the large 2-dimensional adversarial searching space. This is consistent with the empirically observed effectiveness of e2SAD's defense against both FGSM and IFGSM attacks.

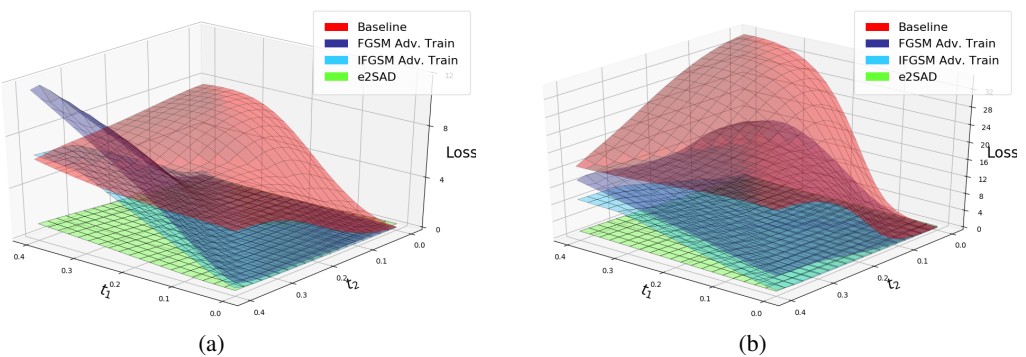

Figure 4: Loss surfaces in a two-dimensional input space of different models: (a) surfaces seen by the FGSM attacks, (b) surfaces seen by the IFGSM attacks.