# OpenReview forum: "EFFICIENT TWO-STEP ADVERSARIAL DEFENSE FOR DEEP NEURAL NETWORKS"
_ICLR.cc/2019/Conference_

### Official Review · AnonReviewer2 · 2018-11-02
**Overall good paper**

**Rating:** 7
**Confidence:** 3

**Review:**

Paper summary: The paper presents a 2-step approach to generate strong adversarial examples at a far lesser cost as compared to recent iterative multi-step adversarial attacks. The authors show the improvements of this technique against different attacks and show that the robustness of their 2-step approach is comparable to the iterative multi-step methods.

The paper presents an interesting technique, is nicely written and easy to read. The fact that their low-cost 2-step method achieves is robust enough to iterative multi-step methods that are expensive is significant.

Pros:
1) The technique is low-cost as compared to other expensive techniques like PGD and IFGSM
2) The technique tries to use the categorical distribution of the generated example in the first step to generate an example in the second step, such that the generated image is most different from the first. This is important and different from the most common technique of iteratively maximizing the loss between the generated samples.
3) The authors show the effetiveness  and improvement of the approach to various attack methods as compared to existing defense techniques
4) The authors evaluate their technique on MNIST and SVHN datasets


Cons or shortcomings/things that need more explanation :
1) It would have been really good to the kind of adversarial examples generated by this technique look like as compared to the examples generated by the other strategies.
2) In table 2, for the substitute models of FGSM trained on H and S labels (rows 2 and 5), it is unclear why the accuracies are so low when attacked on FGSM (hard) and FGSM(soft) models.

---

> ### Public Comment · (anonymous) · 2018-11-09
> **Weak attacks and gradient masking**
>
> FGSM and I-FGSM are not strong attacks and evaluating against them does not mean much - a lot of approaches that claim increased robustness simply cause gradient masking by making the optimization landscape more difficult. The proposed defense has all the hallmarks of gradient masking and will likely break when attacked with PGD with several random restarts.

---

### Official Review · AnonReviewer1 · 2018-11-03
**Missing any theoretical justification, but encouraging empirical result**

**Rating:** 6
**Confidence:** 3

**Review:**

The paper introduces a two-step adversarial defense method, to generate two adversarial examples per clean sample and include them in the actual training loop to achieve robustness. The main claim is that the proposed two-step scheme can outperform more expensive iterative methods such as IFGSM; hence achieving robustness with lower computation. The first example is generated in a standard way (FGSM) method, while the second example is chosen to maximally increase the cross entropy between output distribution of the first and second adversarial example.

The idea seems simple and practical and the empirical results are encouraging. However, other than experiments, there is no justification why the proposed loss should do better that IFGSM. Ideally, I wanted to see the authors to start from some ideal defense definition (e.,g. Eq 4) and then show that some kind of approximation to that leads to the proposed scheme. In the absence of that, the faith about the proposed method solely must be based on the reported empirical evaluation, which is not ideal due to issues like hyper parameter tuning for each of the methods. I hope at least the authors publish the code so it could tried by others.

---

### Official Review · AnonReviewer4 · 2018-11-15
**Interesting research direction but needs more thorough experiments**

**Rating:** 5
**Confidence:** 4

**Review:**

Summary. The authors propose a novel adversarial training method, e2SAD, that relies on a two-step process for generating sets of two training adversarial samples for each clean training sample. The first step is a classical FGSM that yields the first adversarial sample. The second adversarial sample is calculated with a FGSM that is based on the cross-entropy between the probabilities generated by the first adversarial sample and the probabilities generated by the second adversarial sample. The method is computationally efficient (two forward/backward passes per clean sample) w.r.t. powerful iterative attacks such as IFGSM or PGD requiring 40+ steps and the authors claim it gives comparable results to adversarial training with multi-step attacks methods in white and black-box settings.

Clarity. Part 1 and 2 of the paper are well written and summarize the existing attacks/defense mechanisms, their pros and cons as well as the contributions clearly. The next sections could be made shorter (see comments below) to match ICLR’s recommended soft limit of 8 pages instead of the 10 pages hard limit. This would also help the reader grasp the key ideas faster and have a standard formatting (no negative spaces for instance).

Novelty. The idea of simulating the effect of iterative attacks using two distinct steps is novel and appealing to me. The first step increases the loss while the second step shifts the probability distributions apart.

Pros and cons.
(+) The paper is clear and easy to follow, although a bit long.
(+) The idea is interesting and clearly motivated in terms of computational efficiency and in terms of desired properties (Figure 2 illustrates this point well).

(-) Only one aspect of the idea is exploited in the article. It would be interesting to compare this method as an attacker (both in terms of performance and in terms of generated samples, see comment below). Powerful adversarial training should indeed rely on powerful generated adversarial samples.
(-) The results seem somewhat mitigated in terms of significance and conclusions drawn by the authors. Also, the experimental setup is quite light, notably the used CNN architectures are quite small and other datasets could have been used (also linked to the significance of the results).

Comments.
- Shorter paper. Here are suggested modifications for the paper that could help strengthen the impact of your paper. Section 3.1 could be almost entirely discarded as it brings no new ideas w.r.t sections 1 and 2. Figure 1 summarizes the method well, thus the description in Section 3.2 could be made shorter, especially when displaying Equation (8) right after Figure 1. This would then help reduce the size of Sections 3.2.1 and 3.2.2 (because Equation (8) and Figure 1 would prevent you from repeating claims made earlier in the paper). Algorithm 1 is straightforward and could be placed in Appendix. Conclusion and Result sections could be shortened a little as well (not as much as Section 3 though).

- Significance of the results. The significance of some results is unclear to me. Could the authors provide the standard deviation over 3 or 5 runs? For example, in rows 1, 3, 4, 5, 6 of Table 2, it is not clear it e2SAD performs better than FGSM adversarial training, thus raising the question of the necessity of Step 2 of the attack (which is the core contribution of the paper).

- Experimental setup. The last two rows of Table 1 are encouraging for e2SAD. However, the authors could introduce another dataset, e.g. CIFAR10 or 100 or even ImageNet restricted to 20 or 100 random classes/with fewer samples per class and use deeper modern CNN architectures like ResNets (even a ResNet18). Those models are widely adopted both in the research community and by the industry, thus defense mechanisms that provably work for such models can have a huge impact.

- Defense setup. Is the order of Steps 1 and 2 relevant? What if the authors use only iterations of Step 2?

- Attack setup. Here are a few suggestions for assessing your method in an attack setting: what is the precision of the network, without any defense, given an average dissimilarity L2 budget in the training/test samples, in a white/black box setting? How does it compare to standard techniques (e.g. FGSM, IFGSM, DeepFool, Carlini)? What happens if the authors use their method both for both defense and attack? Could the authors display adversarial samples generated by their method?

Conclusion. The idea presented in the paper is interesting, but (1) the experimental results are not entirely satisfactory for the moment and (2) only one aspect of the idea is exploited in the paper, which can be made more interesting and impactful while studying both attack and defense setups. I strongly encourage the authors to continue their research in this area due to the high potential impact and benefits for the whole community.

---

### Public Comment · (anonymous) · 2018-10-25
**Table 3 Concerns**

In Table 3, when attacking e2SAD with IFGSM at 10 iterations you degrade the model to 33% accuracy and at 20 iterations you degrade the model to just 40% accuracy. Is this a statistically significant difference? If it is, this is concerning because more iterations of an attack should produce only stronger results.

---

### Public Comment · (anonymous) · 2018-10-29
**Gradient Masking?**

Lots of recent defenses have been shown to be causing gradient masking. Given that you are performing adversarial training similar to Tramer et al. (2018), which is not effective in a white-box setting, do you have evidence your defense is not just performing gradient masking? Athalye et al. (2018) suggest a few tests for this, such as trying random noise, or using many iterations of PGD.

---

### Meta-Review · Area_Chair1 · 2018-12-17
**Needs improvement.**

**Confidence:** 4
**Recommendation:** Reject

**Metareview:**

While the proposed method is novel, the evaluation is not convincing. In particular, the datasets and models used are small. Susceptibility to adversarial examples is tightly related to dimensionality. The study could benefit from more massive datasets (e.g., Imagenet).